# Age and Social History Impact Social Interactions between Bull Asian Elephants (*Elephas maximus*) at Denver Zoo

Taylor S. Readyhough [1,2,3,*], Maura Davis [4], Sharon Joseph [2,5], Anneke Moresco [2,6] and Amy L. Schreier [1,2]

1   Department of Biology, Regis University, Denver, CO 80221, USA
2   Department of Animal Welfare and Research, Denver Zoo, Denver, CO 80205, USA
3   Department of Natural Resources and the Environment, University of New Hampshire, Durham, NH 03824, USA
4   Department of Animal Care, Denver Zoo, Denver, CO 80205, USA
5   Birmingham Zoo, Birmingham, AL 35223, USA
6   Reproductive Health Surveillance Program, Morrison, CO 80465, USA
*   Correspondence: taylor.readyhough@unh.edu

**Abstract:** Wild bull Asian elephants spend time in all-male groups. Therefore, managers of ex situ populations increasingly house bulls together. We examined the social interactions of five bull Asian elephants at Denver Zoo, using instantaneous sampling to compare social interactions across adolescent and mature bulls, and bulls with a social history prior to the integration of this group compared to bulls with no social history. Both age and social history significantly affected bull behavior. Adolescent bulls exhibited more affiliative and submissive behaviors when housed with mixed-age and mature social partners compared to with only adolescents, and less non-contact agonistic behavior and less time in proximity to a conspecific with mixed-age groups compared to with only other adolescents. Mature bulls exhibited more affiliative behavior when they were with only adolescent bulls compared to only mature bulls, and more time in proximity to a conspecific and increased contact agonistic behavior with at least one adolescent compared to only mature bulls. Bulls in new social groups engaged in more affiliative, agonistic, and submissive behaviors, and spent less time in proximity, than when they were in previously established social combinations. As more institutions house bulls socially, our results provide insights into factors that may affect bull social interactions.

**Keywords:** animal welfare; social behavior; group compatibility; elephant management; pachyderms

## 1. Introduction

In polygynous mammals, male social relationships tend to be competitive as males vie for access to breeding opportunities [1,2]. Strong male associations are rare and are most common in species with male philopatry [2]. However, most polygynous mammals also exhibit sexual segregation which allows all-male groups to form for portions of the year [2–7]. Associating in all-male groups may offer various benefits to males including protection from predators [2,8,9], group defense of high-value resources or female groups [2,3,5,7,10], and shared ecological knowledge of foraging opportunities [2,11].

Historically, mature bull elephants were considered completely solitary except when they were associating with female herds for breeding purposes [12–14]. However, recent research provides clear evidence that bull elephants are more social than previously reported and that they often spend time in all-male groups [4,11,15–18]. Bull African elephants form all-male groups ranging from 2–40 individuals [17], with the size of these groups varying seasonally, often in relation to resource abundance [19]. Additionally, there is some evidence that bull Asian elephants form all-male groups in high-risk environments, including human-dominated landscapes, or to undertake high-risk activities such as crop-raiding or

crossing roadways [16,18,20]. This collaboration may reduce individual risk and increase survival [18,20].

Additional research indicates that the amount of time bulls spend alone, with female herds, and in bull groups varies throughout their lifetime [4,21,22]. Young bulls are forced out of their natal herds between the ages of 9 and 15 years (African elephants 10–15 years, [23]; Asian elephants 9–15 years, [15]). These bulls often continue to occasionally associate with their natal herds as well as with all-male groups [23–25]. Adolescent bulls (ages 11–20 years) spend more time alone and in all-male groups as they reach sexual maturity with the onset of musth and become completely independent from their natal herds [24,26,27]. These adolescent bulls often preferentially associate with older males to learn important social and ecological information [25,27,28], and with similarly-aged conspecifics as they practice sparring and other social interactions through play [25,28–30]. In African elephants, younger males follow older bulls to observe reproductive behaviors, explore novel foraging grounds, and undertake risky behaviors, especially in human-dominated landscapes [11,28]. Bull Asian elephant sociality is understudied due to many challenges including locating and following animals in forest habitats [14,31]. However, groups of Asian bulls who raid crops are often led by an older male, indicating that some behaviors are passed down to younger males through social learning [18,20,32].

Mature male elephants undergo an annual musth cycle marked by increased testosterone production, decreased foraging, and increased roaming and aggression [24,27–29,33,34]. Musth appears to reduce direct competition with other males and increase mating opportunities [35]. Musth is asynchronous in elephant populations, and in African elephants the presence of an older musth bull can suppress musth in younger males [34–37]. In South Africa, introducing mature bulls suppressed musth in adolescent males and decreased adolescent aggression towards white rhinos [37]. Musth also directly affects bull elephant social groups; most often, musth bulls are solitary or associate with female herds instead of with other bulls [27,29].

Beyond patterns of association based on age and reproductive status, males may be more likely to associate with closely related conspecifics with whom they share a social history [3,4,10,38]. In many species, bachelor groups consist of closely related males and the increased survival of any individual is a form of kin selection (e.g., lions, [3]; chimpanzees, [38]; and bottlenose dolphins, [7]). In elephants, younger bulls may socialize with more closely related bulls as they disperse from the same natal herd, and these associations can last throughout their lifetimes [4]. Additionally, in long-lived species, older males may pass knowledge on to related males through association in a form of kin selection, similar to the role of grandmothers in both elephant and orca whale societies [4,39–41].

Elephant groups engage in both affiliative and agonistic social interactions, and all-male groups likely exhibit more agonistic behaviors than female herds [42]. Affiliative behaviors in elephants include maintaining body contact, extending their trunk towards conspecifics, and playing, while agonistic behaviors can include both contact (e.g., sparring, kicking, pushing) and non-contact interactions (e.g., displaying, supplanting, charging) [16,43]. Agonistic interactions are rare even between bulls [16] and often decrease over time after the introduction of a novel animal into an established group [44]. Generally, animal managers consider high amounts of affiliative behavior and low amounts of agonistic behaviors indicators of good animal welfare [45–47].

Historically, male elephants in managed care were housed alone except when they were temporarily placed with females for breeding purposes [23,48,49]. However, with the growing body of evidence for male elephant sociality, institutions increasingly house their bull elephants in social groups [23,48,49]. While some studies have documented the process of introducing a novel bull into an existing group [44,50–52], no studies that we are aware of have examined how individual bulls' age or social history impact social interactions with other bulls during the process of social integration. As more institutions house bulls together, these shifts in management practice require a deeper understanding of the effects of individual bulls' age and social histories to support management decisions.

We hypothesize that age impacts bull elephants' social behavior. Specifically, we predict that adolescent bulls will engage in less contact and non-contact agonistic behaviors and spend more time engaging in affiliative and submissive behaviors, and less time in proximity, with mature conspecifics compared to other adolescents with whom they compete for similar positions in the social network. We expect that mature bulls will engage in more contact and non-contact agonistic behaviors and spend less time on affiliative and submissive behaviors, and less time in proximity, with adolescent conspecifics than with mature conspecifics in order to establish dominance.

We also hypothesize that previous social experience influences bull behavior. We predict that bulls will exhibit more affiliative behavior, spend more time in proximity, and engage in fewer contact agonistic, non-contact agonistic, and submissive behaviors with known conspecifics from previous social groups than with novel social partners as they establish a new social network.

## 2. Materials and Methods

### 2.1. Study Population and Study Site

We examined the social behaviors of five bull Asian elephants housed socially at Denver Zoo, Denver, CO, USA, from February 2019 through the first week of January 2020 [50–52]. The elephant facilities consist of 1.09 hectares including five outdoor yards with varied terrain and pools, as well as eight indoor stalls and a larger indoor parlor. Keepers rotated elephants among the outdoor yards and indoor stalls daily based on scheduled social combinations, weather, and husbandry activities [50–52].

Three unrelated bulls (Individuals 1, 2, and 3; 11 yo, 14 yo, and 49 yo at outset of the study) were sometimes housed together beginning in 2016 [50–52]. They were joined by two half-brothers (Individuals 4 and 5; 9 yo and 10 yo) in September 2018. For the purposes of this study, we considered Individuals 1, 4, and 5 to be adolescents as they did not consistently undergo musth, while Individuals 2 and 3 were considered sexually mature as they entered musth each year. In February 2019, keepers provided the two new bulls three days of "howdy" contact (i.e., auditory, olfactory, visual, and tactile contact through vertical stall bollards) with the other bulls before beginning unrestricted physical introductions. Within one week, all bulls were introduced to one another and were gradually housed in social combinations of 2–5 elephants over the next five months; social combinations included various pairs, trios, and quads, as well as all five elephants. For portions of the five-month introduction period, Individuals 1, 2, and 3 were in musth and, therefore, housed alone [50–52].

All initial introductions between elephants occurred during the daytime in an outdoor yard (0.135 hectares) that had two separate entrances into the indoor elephant barn and access to another yard to provide ample space for elephants to choose to interact with each other and for ease of separating the elephants if intervention was required [50–52]. All introductions were monitored by multiple animal care staff and via video camera. Introductions were initially kept short (30–60 min), and the duration gradually increased to include full days in a certain social combination. Veterinarians and animal care staff agreed to separate the bulls if one became significantly injured or stressed during introductions, although this was never necessary. After the five-month introduction period, social unit size and composition varied through the rest of the study period depending on musth, stage of group integration, and husbandry needs [50–52].

Animal care staff sometimes housed the original three bulls (Individuals 1, 2, and 3) together in various combinations overnight throughout our study, excluding individuals in musth [50–52]. The two new elephants (Individuals 4 and 5) were frequently housed together overnight to maintain consistency as they transitioned from their previous institution where they were always housed socially with their family group. Animal care staff began occasionally housing all five bulls in various social combinations overnight in August 2019, after they determined that the daytime social groupings were stable and included very limited agonistic behavior. Throughout this study, bulls were sometimes housed

alone, especially when they were in musth or needed to be housed alone for husbandry procedures, and often overnight [50–52].

*2.2. Data Collection*

From February 2019 through the first week of January 2020, we used instantaneous scan sampling [53] to record the elephants' behaviors every minute over 30-min samples when they were housed socially during the day [50–52]. Additionally, we used video recordings to observe the elephants overnight when they were housed socially, collecting data following the same methods. The two daytime data collection periods were between 9:30 and 11:30 and between 13:30 and 15:30, which coordinated with the keepers' and elephants' schedules so that we could collect data when there was no elephant–keeper interaction. We conducted all observations from visitor viewing areas so that we would not interfere with typical behavior. Nighttime data collection occurred via video recordings on a rotational schedule for one hour between 18:00 and 20:00, one hour between 21:00 and 23:00, one hour between 0:00 and 2:00, and one hour between 3:00 and 6:00, for a total of four hours of data per night. Both daytime and nighttime observations spanned 4–5 days per week including both weekdays and weekends. Daytime observations were conducted by a team of seven researchers, with the majority of the observational data collected by three individuals. Researchers used daytime practice videos and simultaneous observation sessions to train for data collection. Nighttime videos were processed by two of the researchers who collected most of the daytime data and trained graduate student observers. The researchers instructed observers using video recordings and everyone practiced with simultaneous observations of the same videos. After training, researchers and observers simultaneously scored elephant behaviors during live, daytime observations and recorded nighttime videos, and achieved a 95% inter-observer reliability rate (95% agreement across all behaviors scored at the same time from simultaneous observation sessions of the same focal animal) prior to beginning formal data collection [50–52].

We categorized social behaviors as affiliative, contact agonistic, non-contact agonistic, or submissive ([50–52]; Table 1). When the focal animal was directly interacting with a conspecific during a scan, we recorded the identity of the social partner. We used a mobile application, Zoomonitor®, to collect all behavioral data (Lincoln Park Zoo and Zier Niemann Consulting, 2018). Denver Zoo's Research Committee and Animal Welfare Committee reviewed and approved the study protocol (DZ#2018-008) [50–52].

**Table 1.** Ethogram of behaviors for bachelor group of five bull Asian elephants at Denver Zoo.

| Behavior Category | | Behavior | Definition |
|---|---|---|---|
| Agonistic | Non-Contact | Approach head high | Actor moves toward recipient to within two body lengths with head above shoulders and ears out perpendicular |
| | | Charge | Rapid forward lunging or rapid gait by actor towards a stationary conspecific starting from more than two body lengths away |
| | | Chase | Actor rapidly pursues recipient, who is moving away from actor |
| | | Head shake | Actor holds head above shoulders and moves vigorously from side to side, up and down, or in circular motion |
| | | Supplant | Actor approaches to within two body lengths of conspecific without making contact, causing recipient to turn away or yield ground |
| | Contact ** | Grasp tail | Actor places tail of conspecific into its own trunk while recipient attempts to move away from focal animal |
| | | Kick | Actor strikes at recipient with rear limb |
| | | Mount | Actor rears up on hind legs and places forelegs on recipient |
| | | Push | Actor contacts conspecific with enough force to displace recipient |
| | | Spar | Two elephants mutually and simultaneously push one another backwards with force with heads and/or heads and trunks |
| | | Trunk over back | Actor places 2/3 or more of its trunk firmly over the back or head of a conspecific |

**Table 1.** *Cont.*

| Behavior Category | Behavior | Definition |
|---|---|---|
| Affiliative | Approach relaxed | Actor moves to within to within two body lengths of recipient with head low and ears lying flat against its head, not associated with any other behavior |
| | Body contact | Body contact unspecified in any other behavior (e.g., side-to-side rubbing or touching) |
| | Play | Actor voluntarily spars, wrestles with, mounts, or chases recipient without obvious intent to do harm or display dominance; does not include when following agonistic interaction |
| | Shares food/object | Actor either feeds or uses an object in concert with another elephant that is within one body length |
| | Trunk tangle | Actor loosely entwines its trunk with that of recipient |
| | Trunk to mouth | Actor places its trunk in another elephant's mouth |
| | Trunk touch/toward | Actor extends trunk toward recipient with or without touching; not associated with any other behavior |
| Submissive | Allow | Actor remains still and calmly permits physical contact by conspecific, including genital investigation |
| | Back into/toward | Actor takes two steps (minimum) backward towards another elephant to within one body length, with or without touching |
| | Lower head or ears | Actor quickly drops head and/or ears in response to approach by another elephant |
| | Run away | Actor flees from conspecific in response to its agonistic contact, display, or approach |
| | Turn away/yield | Actor turns body away from or yields ground as a result of actions or encroachment by another elephant |
| Other | Bathe/swim | Actor lies, stands, or submerges in pool (includes spraying water on self); not associated with any other ethogram behavior |
| | Drink | Actor uses trunk to bring water to its mouth and drink |
| | Dust/mud | Actor uses trunk to throw dirt, sand, shavings, or mud onto body while standing |
| | Enrichment interaction | Actor interacts with provided non-food enrichment items |
| | Feed | Actor ingests presented diet items; includes manipulating food items |
| | Follow | Actor closely trails behind recipient, who is moving away from actor (at normal walking speed) |
| | Genital investigation | Actor sniffs or touches genitals of another elephant with its trunk |
| | Locomotion | Actor moves directionally along a horizontal surface (not while feeding); can include slow or fast walking or running |
| | Rest | Stationary; lying down or standing with trunk resting loosely on the ground; eyes open or closed; not performing any other behavior |
| | Stereotypy | Actor performs stereotypic behavior including head-bobbing or pacing |
| | Wallow | Actor lies or rolls in mud or dirt |
| | Other | Actor performs any behavior not on ethogram |
| Out of View | Out of view | Actor cannot be seen or cannot be distinguished from other elephants |

** We included Displaced aggression, Knock-down, and Trunk slap in our data collection protocol as we were interested in recording severe aggression during introductions. However, we never observed any of these behaviors during the instantaneous sampling of elephants.

### 2.3. Data Analysis

We collected a total of 796 h of focal observations (1592 30-min samples) when elephants were housed with at least one conspecific, including 269 h (538 30-min samples) during the daytime and 527 h (1054 30-min samples) during the night. We conducted separate analyses on adolescent and mature focal animals as we expected that younger and older bulls would respond differently to their social group. We categorized *Social-Group* regardless of the focal animal's age; therefore, for analyses, a group was labelled as adolescent if all conspecifics excluding the focal individual were adolescents, mature if all bulls except for the focal animal were mature, and mixed-age if the bulls (excluding the focal animal) were a mixture of adolescent and mature bulls. We converted behavioral categories into binary variables (1 = behavior category occurred; 0 = a different behavior category occurred; Table 2) for each interval scan using the dplyr package in R [54]. We used binomial Generalized Linear Mixed-Effect Models (GLMER) with a logit link to model behavioral responses to various predictors of interest via the *lme4* package in R [55]. In order to test our predictions, we compared (1) odds of engaging in each behavior category (affiliative, non-contact agonistic, contact agonistic, submissive, and in proximity) between the focal animal's age group (adolescent = 9, 10, and 11 yo; mature = 14 and 49 yo) and (2) odds of engaging in each behavior category (affiliative, non-contact agonistic, contact agonistic, submissive, and in proximity) depending on the focal animal's social history (1 = novel social group; 0 = social group established prior to 2018).

**Table 2.** Description of variables and interactions considered in our Generalized Estimating Equation Models (GEE).

| Variable | Description | Reference Level |
|---|---|---|
| Affiliative | Binary variable indicating if the focal animal was engaging in affiliative behavior (0 = no, 1 = yes) | NA—response variable |
| ContactAg | Binary variable indicating if the focal animal was engaging in contact agonistic behavior (0 = no, 1 = yes) | NA—response variable |
| NonContactAg | Binary variable indicating if the focal animal was engaging in non-contact agonistic behavior (0 = no, 1 = yes) | NA—response variable |
| Submissive | Binary variable indicating if the focal animal was engaging in submissive behavior (0 = no, 1 = yes) | NA—response variable |
| SocialAgeLittles | Categorical variable indicating the composition of the social group (Adolescents, Mixed, Mature) excluding the focal animal | Adolescents |
| SocialAgeOlder | Categorical variable indicating the composition of the social group (Adolescents, Mixed, Mature) excluding the focal animal | Mature |
| AccessArea | Continuous variable indicating the size of the area that the focal animal had access to (per 1000 $ft^2$); 2.00–47.37 | 2000 $ft^2$ |
| InOutAccess | Categorical variable indicating if focal animal had access inside (in), outside (out), or both (both) | Both |
| Musth | Categorical variable indicating which elephant (if any) was in musth during the time of the observation session | None |
| TimeOfDay | Categorical variable indicating if observations took place in the morning, afternoon, or night | Morning |
| IntrosPeriod | Binary variable indicating if the observations took place during the 5 mo introductory period (1) or after (0) | After Introductions |
| NewSocial | Binary variable indicating if the social group included novel individuals (1) or only established social partners (0) | Established |
| InOutAccess*AccessArea | Interaction term between InOutAccess (inside, outside, both) and AccessArea | Both:GroupSize |
| IntrosPeriod*NewSocial | Interaction term between IntrosPeriod (0 = after, 1 = during) and NewSocial (0 = established, 1 = new) | After:Established |
| SessionID | Categorical variable identifying the specific focal session that a scan occurred during | N/A—(used as a random effect) |

The fixed effects we considered for all models included the time period (i.e., five-month introduction period or final six months of study); total area that the focal animal had access to (per 1000 $ft^2$); whether the focal animal had access indoors, outdoors, or both; which bull(s) were in musth; whether the observation session occurred in the morning (9:30–11:30), the afternoon (13:30–15:30), or at night (18:00–6:00; Tables A1–A3). We also tested an interaction term between *AccessArea* and *InOutAccess* in all models as the outdoor yards are much larger than the indoor stalls, and an interaction term between *NewSocial* and *IntroPeriod* as the bulls were likely housed in new social combinations less frequently during the introduction period. All models also included random effects for the observation

session (SessionID) as we expected observations within the same session to be correlated (Table 2). We used hypothesis testing in our model selection process: we first fit a full model and then removed individual fixed-effect terms that were not significant predictors (Tables A1–A3). We then compared the reduced model to the original full model with ANOVA using a Wald test (Tables A1–A3). We did not include data during howdy in our analyses as the elephants did not have full physical access to conspecifics. We used open-source statistical software R [56] and R Studio [57] for all analyses. Values of $p < 0.05$ were considered statistically significant.

## 3. Results

### 3.1. Adolescent Bull Behavior

Age composition of the social group significantly impacted focal adolescent bulls' behavior and their odds of being in proximity to a conspecific. As we expected, adolescent bulls exhibited more affiliative and submissive behaviors in groups with mature males. Specifically, adolescent bulls engaged in affiliative behaviors significantly more in mixed-age groups consisting of both adolescent and mature bulls ($p < 0.001$) and in groups with only older bulls ($p = 0.010$) compared to groups with only adolescents (Figure 1; Table A4). This represents a 223% (95% CI: 144–345%) increase in the odds of adolescents' affiliative behaviors in mixed-age groups and an increase in odds of 148% (95% CI: 110–199%) for adolescents in groups with only mature conspecifics (Table A4). Also as predicted, adolescent bulls engaged in significantly more submissive behaviors in social groups that contained a mature conspecific compared to adolescent-only groups (mature only $p < 0.001$; mixed-age $p < 0.001$), which represents a 311% (95% CI: 224–432%) increase in odds of submissive behaviors with mature bulls and a 271% (95% CI: 156–471%) increase in odds in mixed-age groups (Figure 1; Table A4).

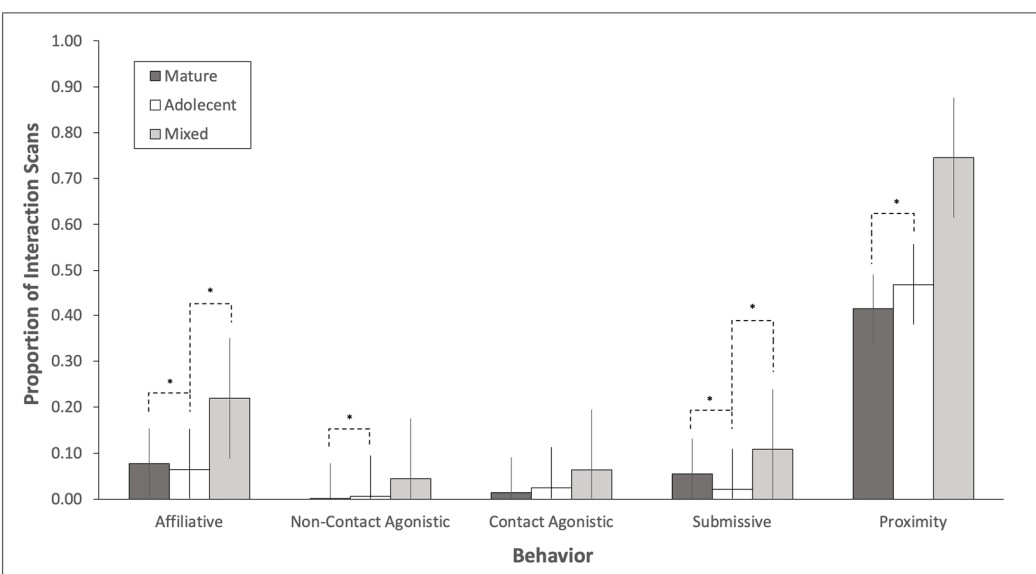

**Figure 1.** Proportion of scans when an adolescent focal animal engaged in social behaviors in different social groups. Adolescent bulls engaged in significantly more affiliative and submissive behaviors in mixed-age and mature groups compared to with only other adolescents. Adolescent bulls also exhibited significantly less non-contact agonistic behaviors and spent significantly less time in proximity to a conspecific when they were housed with mixed-age groups compared to when they were with only other adolescents. Mature = adolescent focal animal housed with group including only bulls who regularly undergo musth; Adolescent = adolescent focal animal housed with group including only other bulls who do not regularly undergo musth; Mixed = adolescent focal animal housed with group including both bulls who do and do not regularly undergo musth. * indicates $p < 0.05$.

Further, adolescent bulls exhibited significantly less non-contact agonism ($p < 0.001$) when housed with mature males compared to when the social group consisted only of adolescent bulls (Figure 1; Table A5). When adolescents were housed with only mature bulls the odds of non-contact agonistic behavior decreased by 82.5% (95% CI: 60.3–92.2%). Unexpectedly, there was no significant change in adolescent bulls' agonism when they were housed in mixed-age groups containing both mature and adolescent conspecifics versus with other adolescents only (non-contact $p = 0.338$; contact $p = 0.947$) and no significant difference in adolescent bulls' contact agonism when they were housed only with older bulls ($p = 0.101$; Table A5; Figure 1). Contrary to our predictions, adolescent bulls were in proximity to a conspecific significantly less when they were housed with only mature bulls ($p < 0.001$), but there was no significant difference when they were housed in mixed-age groups ($p = 0.186$; Table A6; Figure 1) compared to when they were housed only with other adolescents. This represents a 57.9% (95% CI: 35.3–72.6%) decrease in the odds of being in proximity of mature bulls (Table A6).

### 3.2. Mature Bull Behavior

The two mature bulls did not exhibit significant differences in behaviors in mixed-age social groups consisting of adolescent and mature conspecifics compared to when they were housed only with the other mature male (Tables A7 and A8). Further, mature males did not significantly alter their non-contact agonistic or submissive behaviors in different social groups (Tables A7 and A8; Figure 2). However, the mature bulls engaged in significantly more affiliative and contact agonistic behaviors when they were housed with only adolescent bulls versus with a mature conspecific (affiliative $p = 0.037$; contact agonistic $p = 0.019$; Tables A7 and A8; Figure 2). This change represents a 217% (95% CI: 105–451%) increase in the odds of mature bull affiliative behavior and a 514% (95% CI: 131–2011%) increase in contact agonistic behavior with adolescent social partners (Tables A7 and A8; Figure 2). Mature bulls also increased their contact agonistic behaviors when they were housed with mixed-age social partners (985% increase; 95% CI: 165–5880%; $p = 0.012$; Table A8; Figure 2). Interestingly, mature bulls were in proximity to a conspecific significantly more when they were housed with adolescent bulls only ($p < 0.001$) and when they were housed in mixed-age groups with both adolescent and mature social partners ($p = 0.017$; Table A9; Figure 2) compared to when they were housed with only the other mature bull. The odds of mature bulls being in proximity to a conspecific increased by 610% (95% CI: 214–1738%) when they were housed with only adolescent bulls and increased by 701% (95% CI: 142–3464%) when they were housed with both younger and mature bulls (Table A9; Figure 2).

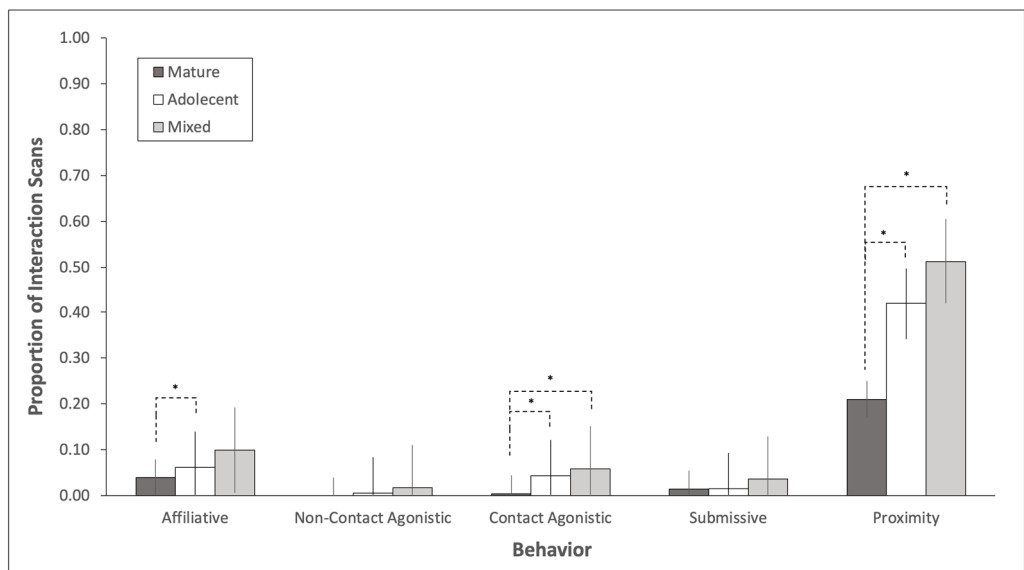

**Figure 2.** Proportion of scans when a mature focal animal engaged in social behaviors in different social groups. Mature bulls exhibited significantly more affiliative behavior when they were housed with only adolescent bulls compared to when they were with only other mature bulls. Mature elephants engaged in more contact agonistic behavior and spent significantly more time in proximity to a conspecific in social groups that included only adolescent bulls and in mixed-age social groups compared to when they were housed with only the other mature bull. Mature = mature focal animal housed with group including only the other bull who regularly underwent musth; Adolescent = mature focal animal housed with group including only bulls who do not regularly undergo musth; Mixed = mature focal animal housed with group including both bulls who do and do not regularly undergo musth. * indicates $p < 0.05$.

### 3.3. Social History Effects on Behavior

Social history decreased the bulls' odds of exhibiting social behavior (affiliative, contact agonistic, non-contact agonistic, and submissive) compared to interactions with novel social partners. We observed significant increases in affiliative behavior ($p = 0.029$), agonistic behaviors (non-contact $p < 0.001$; contact $p < 0.001$), and submissive behavior ($p < 0.001$) in novel social groups compared to groups with an established social history (Tables A10 and A11; Figure 3). This represents an increase in the odds of 134% (95% CI: 103–175%) for affiliative behavior, 352% (95% CI: 170–726%) for non-contact agonism, 665% (95% CI: 380–1162%) for contact agonism, and 367% (95% CI: 247–546%) for submissive behavior in novel social groups compared to established social groups (Tables A10 and A11; Figure 3). When we control for covariates, the odds of the bulls being in proximity to a conspecific also significantly increased by 210% (95% CI: 131–335%; $p = 0.002$) in novel social groups compared to groups with social history (Table A12; Figure 3).

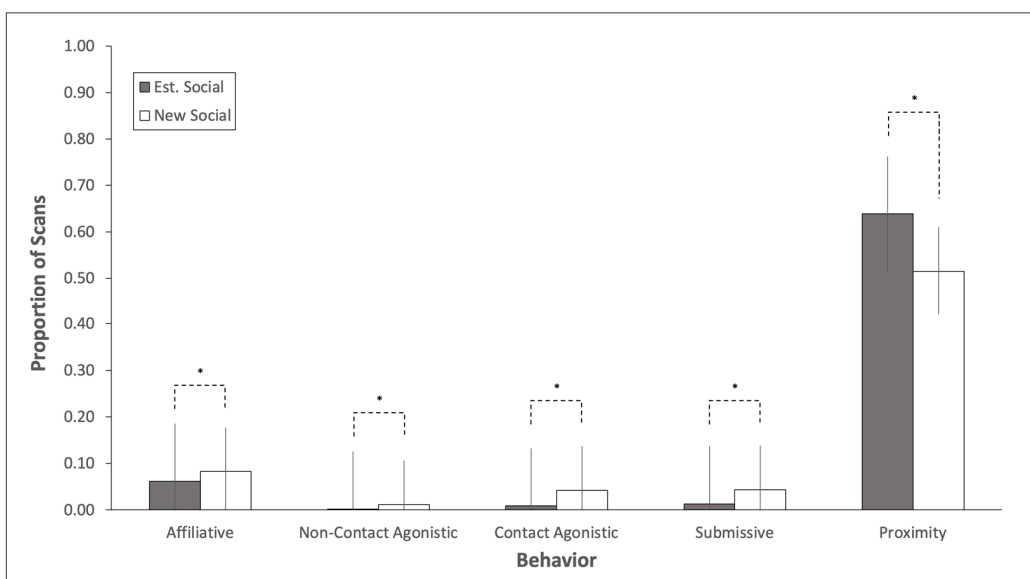

**Figure 3.** Proportion of scans when a focal animal engaged in social behaviors in different social groups. Bulls housed in new social groups engaged in significantly more affiliative behavior, more agonism (both non-contact and contact), and more submissive behaviors than when they were in established social groups. Despite the increase in some social behaviors, bulls spent significantly less time in proximity to a conspecific in new social groups compared to established social groups. Est. Social = social groups that were housed together prior to this study, both at Denver Zoo and at Individual 4 and 5′s previous institution. New Social = social groups introduced during this study, beginning in February 2019. * indicates $p < 0.05$.

## 4. Discussion

Overall, bull elephants' age and social history impacted their social behavior and time in proximity, but not always in ways we predicted. With respect to our first hypothesis that age impacts bull elephants' social behavior, the presence of a mature bull significantly affected adolescent bulls' behavior, and the presence of adolescent bulls significantly altered mature bulls' affiliative and contact agonistic behavior. As expected, adolescent bulls engaged in fewer agonistic behaviors and more affiliative and submissive behaviors with mature conspecifics than with only adolescent conspecifics. This pattern of increased affiliative behaviors and decreased submissive behaviors remained when focal adolescent bulls were in mixed-age groups consisting of mature and adolescent conspecifics, but there was no significant difference in the focal adolescents' agonistic behavior in mixed-age groups. Including at least one mature male in social groups may temporarily reduce competition between adolescents, resulting in the increased affiliative behaviors that we observed in groups containing at least one mature bull. As more institutions house bull elephants together, animal managers can use this understanding of expected amounts of affiliative, agonistic, and submissive behaviors to inform which individuals to include in social groupings, especially during the introductory period. Specifically, managers might begin the introduction process by focusing on combining mature and adolescent bulls in small social groupings to promote low levels of agonism, while delaying housing groups together that only contain adolescent bulls as these groupings displayed the highest levels of agonism in our study. Further, empirically measuring rates of affiliative, agonistic, and submissive behavior in different social groupings empowers animal managers to choose to house bulls together based on specific measurements which could be compared to this study of the successful integration of two bull groups.

Mature bulls altered their behavior in different social groups, but the shifts were mostly contrary to our predictions. Interestingly, there was a significant increase in mature bulls' affiliative behavior when they were housed with only adolescent bulls, and a similar increase in contact agonistic behavior in mixed-age groups, which had both mature and

adolescent conspecifics, compared to groups with only the mature bulls. Unexpectedly, mature bulls spent more time in proximity with conspecifics when they were housed with at least one adolescent bull, compared to with the other mature male. Overall, the consistency of mature bulls' behavior across social groups could indicate that bulls who reliably undergo musth hold more stable positions in bull social networks. For example, in wild African elephant populations older males held more central positions in social networks [4]. Furthering our understanding of this social stability allows animal managers to anticipate bulls' interactions and may indicate that mature bulls can mediate agonistic interactions between younger conspecifics.

For this study, we considered mature bulls to be animals who consistently underwent an annual musth period, which included Individual 2 (14 years old at the beginning of this study) and Individual 3 (49 years old at the beginning of this study). Both mature individuals were physically larger than the adolescent bulls; however, Individual 2 was still growing and while he underwent a musth period annually, the duration and timing of that period varied from year to year. The "mature bull" category covered a very large age range, and it is possible that our analysis did not capture more nuanced social interactions due to the discrepancy in age between the two mature males. Few institutions currently house Asian elephant bull groups, so it is difficult to ensure that bachelor groups include more than one mature adult of similar ages [48,49]. We are hopeful that as more facilities house multiple bulls, opportunities for socializing diverse age groups will increase, allowing for more bull groups that consist of multiple mature adults as well as multiple adolescent bulls. Anecdotally, the presence of the oldest bull (Individual 3) exerted a much larger influence on the adolescent bulls' behavior than the presence of the newly mature male (Individual 2); the eldest bull directly interrupted sparring matches between adolescents on several occasions while we never observed the newly mature male actively intervening (Davis pers. comm.). Further investigation into the effects of a newly mature male versus a fully mature or geriatric male would elucidate the influence of older, dominant bulls on adolescent conspecifics.

Regarding our second hypothesis that bull elephants' social history affects social behaviors, and as we expected, an established social history resulted in significant decreases in the odds of agonistic and submissive behaviors and a significant increase in the odds of being in proximity to a conspecific compared to when bulls were interacting in novel social combinations. These results indicate that at this stage of social integration, bulls may experience more social stress with new partners compared to established social partners. This difference may also represent a preference for engaging in more social interactions with novel bulls as the group establishes new social bonds and individuals vie for positions within the social network, which is further supported by the unexpected increase in affiliative behaviors that we observed in new social groups. As more facilities house bull Asian elephants together, animal managers can build upon previously established social bonds between bulls, especially during the stressful introductory period. For example, animal managers could group two bulls with an established positive social history together with a novel individual, especially when the newcomer is older or physically larger than the two established bulls.

This study ended approximately 11 months after Individuals 4 and 5 were first physically introduced to Individuals 1, 2, and 3. It is possible that it takes longer than that for such long-lived mammals to establish stable social relationships especially when socialization of mature bulls is decreased due to separation during periods of musth. For example, a study investigating the integration of a new adolescent bull into a bachelor group at Heidelberg Zoo found that the new bull engaged in more play and participated in more friendly interactions one year after introductions compared to the four months after his initial introduction [44]. Future studies should compare groups' social behaviors close to introduction and a few years later to expand our understanding of longer-term social dynamics between bull Asian elephants.

In range countries, adolescent bull elephants often associate with their natal herds and with other closely related young bulls [4,25,28]. These social preferences can continue throughout the bulls' lives, and some studies indicate that bulls are more likely to associate with closely related bulls throughout their lifetimes [4]. These patterns of preferential association may support the kin selection hypothesis, whereby animals share ecological knowledge with closely related conspecifics to increase overall fitness [58]. However, it is important to note that we cannot empirically separate a long social history and genetic relatedness in these cases. This study indicated that bull Asian elephants engaged in fewer agonistic and submissive behaviors with, and spent more time in proximity to, bulls with whom they had an established social history compared to novel social partners. We hypothesize that a long social history may support the close social bonds observed between related individuals in wild populations and serve to benefit the inclusive fitness of both animals by reducing conflict between related individuals.

Other variables in our models that significantly affected the bulls' social interactions included time of day, access inside or outside, and introductory period. Bulls engaged in all social behaviors more often during the morning observation period compared to the afternoon or nighttime observation periods (Appendix B). Bulls were also more socially active when they had access to both inside stalls and outdoor yards than when they were housed only indoors (Appendix B). Previous studies indicated that more than exhibit size, social management and complexity can influence elephant welfare, and the enrichment provided from social interaction may improve animal welfare at institutions where exhibit size is limited [48,59,60]. The bulls' social behavior differed during the introductory period (the first 5 months after introductions began) compared to later in our study; generally, the bulls engaged in less social behavior during the introduction period compared to after the first five months (Appendix B). Previous studies of both bull and cow introductions indicated that the initial introductory period is a time of increased stress, but that this stress decreases over time after introductions [44,61], which may account for the increased social behavior we observed once the bulls had at least five months of social interaction.

During this study we did not house the mature bulls socially when they were in musth or the adolescent bulls when they showed signs of moto-musth (e.g., temporal gland drainage; unprompted aggression towards facilities, conspecifics, or staff). However, musth bulls were still housed in the same building and in neighboring outdoor yards, so all bulls had auditory and olfactory access to an animal during musth. There is strong evidence for musth suppression of adolescent bulls in African elephants [28,34,37], but much less is understood about the effects of musth in Asian elephant bull groups [24,35,36]. The potential for concurrent or overlapping musth periods with Asian elephant bulls further complicates housing multiple bulls at one facility. Opportunities for future studies on the effects of Asian elephant sociality on musth continue to increase as more institutions house bulls together, and as some institutions focus on solely housing bull elephants.

While this study only included five Asian elephants at one facility, we provide an initial investigation into the effects of age and social history on bulls' social interactions. As more institutions house bulls socially, we hope our results provide insights into some of the demographic and life-history factors that may affect bull social interactions during the introductory period and beyond. Improving animal welfare is our "duty of care" [62] and as animal managers we can no longer disregard the evidence of bull elephant sociality [48,49,63]. Bull elephants often exhibit more agonistic behaviors than females [42], leading to common, and well-founded, concerns about socializing bulls in managed care. This study and similar investigations into bull elephant sociality prepare animal managers to plan species-appropriate social housing and support positive animal welfare, especially as populations of elephants in range countries face unprecedented changes and ex situ populations become increasingly valuable for conservation [64].

**Author Contributions:** Conceptualization, S.J., A.M. and A.L.S.; methodology, T.S.R. and A.L.S.; software, T.S.R.; validation, A.L.S.; formal analysis, T.S.R.; investigation, S.J., M.D., T.S.R. and A.L.S.; resources, S.J., M.D. and A.M.; data curation, T.S.R.; writing—original draft preparation, T.S.R.;

writing—review and editing, S.J., M.D., A.M. and A.L.S.; visualization, T.S.R.; supervision, S.J. and A.L.S.; project administration, S.J. and A.M.; funding acquisition, S.J., M.D., A.M. and A.L.S. All authors have read and agreed to the published version of the manuscript.

**Funding:** This research was funded by the Association of Zoos and Aquariums Conservation Grant Fund (grant #18-1523) and by Denver Zoo.

**Institutional Review Board Statement:** The animal study protocol was approved by the Research Committee and Animal Welfare Committee of Denver Zoo (DZ#2018-008, July 2018).

**Data Availability Statement:** The data presented in this study are available on request from the corresponding author. The data are not publicly available due to institutional policy.

**Acknowledgments:** We are thankful for Denver Zoo's elephant care team for their support throughout this project: Lauren Cahill, Sarah Cesler, Rachel Chappell, Barb Junkermeier, Gabe Kibe, Danielle Lints, Jeffery Stanton, and Victoria Wickens. We are also grateful for support from Brian Aucone, Emily Insalaco, Dale Leeds, Courtney Peterson, and Katie Vyas, and for statistical guidance from Kristofor Voss. We also thank Shanelle Thevarajah, Dena Bergman, Dylan Brown, Hannah DeKay, Tyler Erickson, Chelsea Huck, Kaily Meek, Samantha Ortega, Katie Shapiro, Mary Strecker, Dermot Swanson, Richard Patsilevas, Emily Ramos, Rabie Barka, Marley Borham, Bradley Hamilton, Alexandra Sorenson, Alex Stacy, Armando Toral, and Chase Westbrook for nighttime behavior data collection.

**Conflicts of Interest:** The authors declare no conflict of interest. The funders had no role in the design of the study; in the collection, analyses, or interpretation of data; in the writing of the manuscript, or in the decision to publish the results.

## Appendix A. Model Selection Tables for GLMER of Bull Elephant Behavior

**Table A1.** Model selection table for GLMER of adolescent behaviors (adolescent focal data; bold indicates the final model selection). *p*-value from comparison to full model with ANOVA.

| Model | AIC | BIC | X$^2$ | Parameters | *p*-Value |
|---|---|---|---|---|---|
| Affiliative | | | | | |
| SocialAgeLittle + TimeOfDay + Musth + InOutAccess + AccessArea + IntrosPeriod + NewSocial + InOutAccess*AccessArea + IntrosPeriod*NewSocial | 13,768 | 13,908 | - | 17 | - |
| SocialAgeLittle + TimeOfDay + Musth + InOutAccess + AccessArea + IntrosPeriod + NewSocial + InOutAccess*AccessArea | 13,766 | 13,898 | 1.020 | 16 | 0.313 |
| SocialAgeLittle + TimeOfDay + Musth + InOutAccess + AccessArea + IntrosPeriod + NewSocial | 13,763 | 13,878 | 1.432 | 14 | 0.698 |
| SocialAgeLittle + TimeOfDay + Musth + InOutAccess + AccessArea + IntrosPeriod | 13,766 | 13,873 | 6.616 | 13 | 0.158 |
| SocialAgeLittle + TimeOfDay + Musth + InOutAccess + AccessArea | 13,767 | 13,866 | 9.693 | 12 | 0.084 |
| SocialAgeLittle + TimeOfDay + Musth + InOutAccess | 13,762 | 13,853 | 6.701 | 11 | 0.349 |
| **SocialAgeLittle + TimeOfDay + Musth** | **13,765** | **13,839** | **13.428** | **9** | **0.098** |
| Submissive | | | | | |
| SocialAgeLittles + TimeOfDay + Musth + InOutAccess + AccessArea + IntrosPeriod + NewSocial + InOutAccess*AccessArea + IntrosPeriod*NewSocial | 7908.1 | 8048.2 | - | 17 | - |
| SocialAgeLittles + TimeOfDay + InOutAccess + AccessArea + IntrosPeriod + NewSocial + InOutAccess*AccessArea + IntrosPeriod*NewSocial | 7907.0 | 8048.2 | 4.862 | 14 | 0.182 |
| **SocialAgeLittles + TimeOfDay + InOutAccess + AccessArea + IntrosPeriod + NewSocial + InOutAccess*AccessArea** | **7908.5** | **8015.6** | **8.365** | **13** | **0.079** |

**Table A1.** *Cont.*

| Model | AIC | BIC | X² | Parameters | *p*-Value |
|---|---|---|---|---|---|
| *Non-contact Agonistic* | | | | | |
| SocialAgeLittles + TimeOfDay + Musth + InOutAccess + AccessArea + IntrosPeriod + NewSocial + InOutAccess*AccessArea + IntrosPeriod*NewSocial | 2311.4 | 2451.4 | - | 17 | - |
| SocialAgeLittles + TimeOfDay + Musth + InOutAccess + AccessArea + IntrosPeriod + NewSocial + IntrosPeriod*NewSocial | 2309.2 | 2432.8 | 1.808 | 15 | 0.405 |
| SocialAgeLittles + TimeOfDay + Musth + InOutAccess + IntrosPeriod + NewSocial + IntrosPeriod*NewSocial | 2309.6 | 2424.9 | 4.198 | 14 | 0.241 |
| SocialAgeLittles + TimeOfDay + InOutAccess + IntrosPeriod + NewSocial + IntrosPeriod*NewSocial | 2306.1 | 2396.7 | 6.721 | 11 | 0.347 |
| **SocialAgeLittles + TimeOfDay + IntrosPeriod + NewSocial + IntrosPeriod*NewSocial** | **2303.0** | **2377.1** | **7.586** | **9** | **0.475** |
| *Contact Agonistic* | | | | | |
| SocialAgeLittles + TimeOfDay + Musth + InOutAccess + AccessArea + IntrosPeriod + NewSocial + InOutAccess*AccessArea + IntrosPeriod*NewSocial | 5368.2 | 5508.3 | - | 17 | - |
| SocialAgeLittles + TimeOfDay + Musth + InOutAccess + AccessArea + IntrosPeriod + NewSocial + IntrosPeriod*NewSocial | 5367.5 | 5491.1 | 3.274 | 15 | 0.195 |
| **SocialAgeLittles + TimeOfDay + Musth + InOutAccess + IntrosPeriod + NewSocial + IntrosPeriod*NewSocial** | **5365.7** | **5481.0** | **3.481** | **14** | **0.323** |
| *Proximity* | | | | | |
| SocialAgeLittles + TimeOfDay + Musth + InOutAccess + AccessArea + IntrosPeriod + NewSocial + InOutAccess*AccessArea + IntrosPeriod*NewSocial | 25,842 | 25,982 | - | 17 | - |
| SocialAgeLittles + TimeOfDay + Musth + InOutAccess + AccessArea + IntrosPeriod + NewSocial + IntrosPeriod*NewSocial | 25,840 | 25,964 | 1.772 | 15 | 0.412 |
| **SocialAgeLittles + TimeOfDay + Musth + InOutAccess + IntrosPeriod + NewSocial + IntrosPeriod*NewSocial** | **25,839** | **25,954** | **2.180** | **14** | **0.536** |

**Table A2.** Model selection table for GLMER of mature bull behaviors (mature focal data; bold indicates the final model selection). *p*-value from comparison to full model with ANOVA.

| Model | AIC | BIC | X² | Parameters | *p*-Value |
|---|---|---|---|---|---|
| *Affiliative* | | | | | |
| SocialAgeOlder + TimeOfDay + Musth + InOutAccess + AccessArea + IntrosPeriod + NewSocial + InOutAccess*AccessArea + IntrosPeriod*NewSocial | 4392.6 | 4516.2 | - | 17 | - |
| SocialAgeOlder + TimeOfDay + Musth + InOutAccess + AccessArea + IntrosPeriod + NewSocial + InOutAccess*AccessArea | 4390.6 | 4506.9 | 0.011 | 16 | 0.916 |
| SocialAgeOlder + TimeOfDay + Musth + InOutAccess + AccessArea + IntrosPeriod + NewSocial | 4386.7 | 4488.4 | 0.038 | 14 | 0.998 |
| SocialAgeOlder + TimeOfDay + Musth + InOutAccess + AccessArea + IntrosPeriod | 4384.8 | 4479.3 | 0.183 | 13 | 0.996 |
| SocialAgeOlder + TimeOfDay + Musth + InOutAccess + AccessArea | 4387.1 | 4474.3 | 4.430 | 12 | 0.489 |
| SocialAgeOlder + TimeOfDay + Musth + InOutAccess | 4389.5 | 4469.5 | 8.902 | 11 | 0.179 |
| **SocialAgeOlder + TimeOfDay + Musth** | **4387.6** | **4453.0** | **10.961** | **9** | **0.204** |

**Table A2.** *Cont.*

| Model | AIC | BIC | X² | Parameters | *p*-Value |
|---|---|---|---|---|---|
| Submissive | | | | | |
| SocialAgeOlder + TimeOfDay + Musth + InOutAccess + AccessArea + IntrosPeriod + NewSocial + InOutAccess*AccessArea + IntrosPeriod*NewSocial | 1608.7 | 1732.2 | - | 17 | - |
| SocialAgeOlder + TimeOfDay + Musth + InOutAccess + AccessArea + IntrosPeriod + NewSocial + InOutAccess*AccessArea | 1607.2 | 1723.4 | 0.490 | 16 | 0.484 |
| SocialAgeOlder + TimeOfDay + Musth + InOutAccess + AccessArea + IntrosPeriod + NewSocial | 1604.9 | 1706.7 | 2.236 | 14 | 0.525 |
| SocialAgeOlder + TimeOfDay + Musth + InOutAccess + AccessArea + IntrosPeriod | 1607.5 | 1702.0 | 6.851 | 13 | 0.144 |
| SocialAgeOlder + TimeOfDay + Musth + InOutAccess + AccessArea | 1606.3 | 1693.5 | 7.631 | 12 | 0.178 |
| **SocialAgeOlder + TimeOfDay + Musth + AccessArea** | **1606.1** | **1678.7** | **11.393** | **10** | **0.122** |
| Non-contact Agonistic | | | | | |
| SocialAgeOlder + TimeOfDay + Musth + InOutAccess + AccessArea + IntrosPeriod + NewSocial + InOutAccess*AccessArea + IntrosPeriod*NewSocial | 669.6 | 793.1 | - | 17 | - |
| SocialAgeOlder + TimeOfDay + Musth + InOutAccess + AccessArea + IntrosPeriod + NewSocial + IntrosPeriod*NewSocial | 668.7 | 777.7 | 3.157 | 15 | 0.206 |
| SocialAgeOlder + TimeOfDay + Musth + InOutAccess + IntrosPeriod + NewSocial + IntrosPeriod*NewSocial | 667.3 | 769.0 | 3.713 | 14 | 0.294 |
| SocialAgeOlder + TimeOfDay + Musth + IntrosPeriod + NewSocial + IntrosPeriod*NewSocial | 666.7 | 754.0 | 7.168 | 12 | 0.208 |
| SocialAgeOlder + TimeOfDay + IntrosPeriod + NewSocial + IntrosPeriod*NewSocial | 665.8 | 731.2 | 12.261 | 9 | 0.140 |
| SocialAgeOlder + TimeOfDay + IntrosPeriod + NewSocial | 665.9 | 724.1 | 14.343 | 8 | 0.111 |
| **SocialAgeOlder + TimeOfDay + IntrosPeriod** | **664.2** | **715.1** | **14.667** | **7** | **0.145** |
| Contact Agonistic | | | | | |
| SocialAgeOlder + TimeOfDay + Musth + InOutAccess + AccessArea + IntrosPeriod + NewSocial + InOutAccess*AccessArea + IntrosPeriod*NewSocial | 2951.3 | 3074.9 | - | 17 | - |
| SocialAgeOlder + TimeOfDay + Musth + InOutAccess + AccessArea + IntrosPeriod + NewSocial + InOutAccess*AccessArea | 2951.2 | 3067.5 | 1.903 | 16 | 0.168 |
| SocialAgeOlder + TimeOfDay + Musth + InOutAccess + AccessArea + IntrosPeriod + NewSocial | 2948.9 | 3050.6 | 3.519 | 14 | 0.318 |
| **SocialAgeOlder + TimeOfDay + InOutAccess + AccessArea + IntrosPeriod + NewSocial** | **2947.1** | **3027.1** | **7.787** | **11** | **0.254** |
| Proximity | | | | | |
| SocialAgeOlder + TimeOfDay + Musth + InOutAccess + AccessArea + IntrosPeriod + NewSocial + InOutAccess*AccessArea + IntrosPeriod*NewSocial | 10,112 | 10,236 | - | 17 | - |
| SocialAgeOlder + TimeOfDay + Musth + InOutAccess + AccessArea + IntrosPeriod + NewSocial + InOutAccess*AccessArea | 10,110 | 10,226 | 0.172 | 16 | 0.678 |
| **SocialAgeOlder + TimeOfDay + Musth + InOutAccess + AccessArea + NewSocial + InOutAccess*AccessArea** | **10,108** | **10,217** | **0.220** | **15** | **0.896** |

**Table A3.** Model selection table for GLMER of new social group behaviors (all data; bold indicates the final model selection). *p*-value from comparison to full model with ANOVA.

| Model | AIC | BIC | $X^2$ | Parameters | *p*-Value |
|---|---|---|---|---|---|
| *Affiliative* | | | | | |
| NewSocial + TimeOfDay + Musth + InOutAccess + AccessArea + IntrosPeriod + InOutAccess*AccessArea + IntrosPeriod*NewSocial | 18,039 | 18,167 | - | 15 | - |
| NewSocial + TimeOfDay + Musth + InOutAccess + AccessArea + IntrosPeriod + InOutAccess*AccessArea | 18,037 | 18,157 | 0.725 | 14 | 0.395 |
| NewSocial + TimeOfDay + Musth + InOutAccess + AccessArea + IntrosPeriod | 18,037 | 18,140 | 4.216 | 12 | 0.239 |
| NewSocial + TimeOfDay + Musth + InOutAccess + AccessArea | 18,036 | 18,036 | 5.523 | 11 | 0.238 |
| NewSocial + TimeOfDay + Musth + InOutAccess | 18,034 | 18,120 | 5.542 | 10 | 0.353 |
| **NewSocial + TimeOfDay + InOutAccess** | **18,032** | **18,167** | **9.517** | **8** | **0.301** |
| *Submissive* | | | | | |
| NewSocial + TimeOfDay + Musth + InOutAccess + AccessArea + IntrosPeriod + InOutAccess*AccessArea + IntrosPeriod*NewSocial | 9784.3 | 9912.6 | - | 15 | - |
| **NewSocial + TimeOfDay + Musth + InOutAccess + AccessArea + IntrosPeriod + IntrosPeriod*NewSocial** | **9784.2** | **9895.5** | **3.965** | **13** | **0.138** |
| *Non-contact Agonistic* | | | | | |
| NewSocial + TimeOfDay + Musth + InOutAccess + AccessArea + IntrosPeriod + InOutAccess*AccessArea + IntrosPeriod*NewSocial | 3038.2 | 3166.6 | - | 15 | - |
| NewSocial + TimeOfDay + Musth + InOutAccess + AccessArea + IntrosPeriod + IntrosPeriod*NewSocial | 3036.2 | 3147.5 | 2.029 | 13 | 0.363 |
| NewSocial + TimeOfDay + Musth + InOutAccess + IntrosPeriod + IntrosPeriod*NewSocial | 3038.2 | 3140.9 | 6.038 | 12 | 0.120 |
| **NewSocial + TimeOfDay + Musth + IntrosPeriod + IntrosPeriod*NewSocial** | **3036.7** | **3122.3** | **8.537** | **10** | **0.129** |
| *Contact Agonistic* | | | | | |
| **NewSocial + TimeOfDay + Musth + InOutAccess + AccessArea + IntrosPeriod + InOutAccess*AccessArea + IntrosPeriod*NewSocial** | **8390.7** | **8519.1** | **-** | **15** | **-** |
| *Proximity* | | | | | |
| **NewSocial + TimeOfDay + Musth + InOutAccess + AccessArea + IntrosPeriod + InOutAccess*AccessArea + IntrosPeriod*NewSocial** | **35,414** | **35,542** | **-** | **15** | **-** |

## Appendix B. Model Output Tables

This appendix includes detailed model output tables for each of our final models. Results from these outputs are also included in Figures 1–3 in the manuscript.

**Table A4.** Adolescent GLMER model parameter estimates for affiliative and submissive response variables. Generalized linear mixed-effect model results of associations between predictor variables (fixed effects) and behavioral response variables for adolescent focal animals. ß = Beta coefficients from model outputs: positive values indicate an increase in odds compared to the reference level while negative values indicate a decrease in odds compared to reference level; SE = standard error of beta coefficients; Wald $X^2$ = Chi-squared statistic from Wald test with df = 1; $p$ = $p$-value from Wald test. * indicates reference level. Bold signifies statistical significance ($p < 0.05$).

| Predictor | Level | Affiliative | | | | | Submissive | | | | |
|---|---|---|---|---|---|---|---|---|---|---|---|
| | | Odds Ratio | ß | SE | z-Statistic | $p$ | Odds Ratio | ß | SE | z-Statistic | $p$ |
| SocialAge | Adolescent * | - | - | - | - | - | - | - | - | - | - |
| | Mature | **1.479** | **0.392** | **0.152** | **2.582** | **0.010** | **3.078** | **1.124** | **0.155** | **7.252** | **<0.001** |
| | Mixed | **2.226** | **0.800** | **0.224** | **3.574** | **<0.001** | **2.460** | **0.900** | **0.260** | **3.461** | **<0.001** |
| TimeOfDay | Morning * | - | - | - | - | - | - | - | - | - | - |
| | Afternoon | 0.735 | −0.308 | 0.192 | −1.606 | 0.108 | **0.398** | **−0.922** | **0.218** | **−4.234** | **<0.001** |
| | Night | **0.112** | **−2.187** | **0.164** | **−13.309** | **<0.001** | **0.100** | **−2.307** | **0.218** | **−9.125** | **<0.001** |
| InOutAccess | Both * | - | - | - | - | - | - | - | - | - | - |
| | Inside | - | - | - | - | - | 1.679 | 0.518 | 1.157 | 0.448 | 0.654 |
| | Outside | - | - | - | - | - | 0.844 | −0.170 | 1.120 | −0.151 | 0.880 |
| AccessArea | | - | - | - | - | - | 0.999 | −0.001 | 0.064 | −0.009 | 0.993 |
| Musth | None * | - | - | - | - | - | - | - | - | - | - |
| | Indiv. 2 | 1.203 | 0.185 | 0.172 | 1.073 | 0.283 | - | - | - | - | - |
| | Indiv. 3 | **0.653** | **−0.426** | **0.179** | **−2.379** | **0.017** | - | - | - | - | - |
| | Indiv. 2 and 3 | 0.857 | −0.154 | 0.264 | −0.582 | 0.560 | - | - | - | - | - |
| IntrosPeriod | 0 * | - | - | - | - | - | - | - | - | - | - |
| | 1 | - | - | - | - | - | 1.490 | 0.399 | 0.186 | 2.142 | 0.032 |
| NewSocial | 0 * | - | - | - | - | - | - | - | - | - | - |
| | 1 | - | - | - | - | - | **4.460** | **1.495** | **0.173** | **8.636** | **<0.001** |
| InOutAccess*AccessArea | In | - | - | - | - | - | **0.711** | **−0.342** | **0.169** | **−2.017** | **0.044** |
| InOutAccess*AccessArea | Out | - | - | - | - | - | 0.981 | −0.019 | 0.065 | −0.290 | 0.772 |

**Table A5.** Adolescent GLMER model parameter estimates for agonistic response variables. Generalized linear mixed-effect model results of associations between predictor variables (fixed effects) and behavioral response variables for adolescent focal animals. ß = Beta coefficients from model outputs: positive values indicate an increase in odds compared to the reference level while negative values indicate a decrease in odds compared to reference level; SE = standard error of beta coefficients; Wald $X^2$ = Chi-squared statistic from Wald test with df = 1; $p$ = $p$-value from Wald test. * indicates reference level. Bold signifies statistical significance ($p < 0.05$).

| Predictor | Level | No-Contact Agonistic | | | | | Contact Agonistic | | | | |
|---|---|---|---|---|---|---|---|---|---|---|---|
| | | Odds Ratio | ß | SE | z-Statistic | $p$ | Odds Ratio | ß | SE | z-Statistic | $p$ |
| SocialAge | Adolescent * | - | - | - | - | - | - | - | - | - | - |
| | Mature | **0.175** | **−1.741** | **0.416** | **−4.187** | **<0.001** | 0.619 | −0.479 | 0.292 | −1.639 | 0.101 |
| | Mixed | 1.341 | 0.294 | 0.307 | 0.957 | 0.338 | 1.029 | −0.028 | 0.420 | −0.067 | 0.947 |
| TimeOfDay | Morning * | - | - | - | - | - | - | - | - | - | - |
| | Afternoon | **0.496** | **−0.702** | **0.325** | **−2.160** | **0.031** | 0.531 | −0.634 | 0.352 | −1.802 | 0.071 |
| | Night | **0.029** | **−3.538** | **0.472** | **−7.498** | **<0.001** | **0.100** | **−2.307** | **0.411** | **−5.608** | **<0.001** |
| InOutAccess | Both * | - | - | - | - | - | - | - | - | - | - |
| | Inside | - | - | - | - | - | 1.692 | 0.526 | 0.396 | 1.328 | 0.184 |
| | Outside | - | - | - | - | - | 0.841 | −0.173 | 0.376 | −0.460 | 0.646 |
| Musth | None * | - | - | - | - | - | - | - | - | - | - |
| | Indiv. 2 | - | - | - | - | - | 0.663 | −0.412 | 0.351 | −1.174 | 0.241 |
| | Indiv. 3 | - | - | - | - | - | **0.286** | **−1.253** | **0.398** | **−3.146** | **0.002** |
| | Indiv. 2 and 3 | - | - | - | - | - | 0.839 | −0.176 | 0.499 | −0.352 | 0.725 |
| IntrosPeriod | 0 * | - | - | - | - | - | - | - | - | - | - |
| | 1 | 0.871 | −0.138 | 0.685 | −0.201 | 0.840 | 0.742 | −0.298 | 0.521 | −0.572 | 0.568 |
| NewSocial | 0 * | - | - | - | - | - | - | - | - | - | - |
| | 1 | **6.880** | **1.929** | **0.526** | **3.663** | **<0.001** | **5.773** | **1.753** | **0.362** | **4.847** | **<0.001** |
| IntrosPeriod*NewSocial | | **4.386** | **1.478** | **0.746** | **1.983** | **0.047** | **5.577** | **1.719** | **0.588** | **2.922** | **0.003** |

**Table A6.** Adolescent GLMER model parameter estimates for proximity response variable. Generalized linear mixed-effect model results of associations between predictor variables (fixed effects) and proximity response variable for adolescent focal animals. ß = Beta coefficients from model outputs: positive values indicate an increase in odds compared to the reference level while negative values indicate a decrease in odds compared to reference level; SE = standard error of beta coefficients; Wald $X^2$ = Chi-squared statistic from Wald test with df = 1; $p$ = $p$-value from Wald test. * indicates reference level. Bold signifies statistical significance ($p < 0.05$).

| Predictor | Level | Proximity | | | | |
|---|---|---|---|---|---|---|
| | | Odds Ratio | ß | SE | $z$-Statistic | $p$ |
| SocialAge | Adolescent * | - | - | - | - | - |
| | Mature | **0.421** | **−0.865** | **0.219** | **−3.944** | **<0.001** |
| | Mixed | 1.606 | 0.474 | 0.358 | 1.322 | 0.186 |
| TimeOfDay | Morning * | - | - | - | - | - |
| | Afternoon | 0.758 | −0.278 | 0.313 | −0.888 | 0.374 |
| | Night | **0.132** | **−2.028** | **0.319** | **−6.350** | **<0.001** |
| InOutAccess | Both * | - | - | - | - | - |
| | Inside | 0.909 | −0.095 | 0.284 | −0.335 | 0.737 |
| | Outside | **0.362** | **−1.016** | **0.300** | **−3.389** | **<0.001** |
| Musth | None * | - | - | - | - | - |
| | Indiv. 2 | **0.512** | **−0.669** | **0.282** | **−2.375** | **0.018** |
| | Indiv. 3 | **0.288** | **−1.244** | **0.297** | **−4.185** | **<0.001** |
| | Indiv. 2 and 3 | 0.638 | −0.449 | 0.416 | −1.078 | 0.281 |
| IntrosPeriod | 0 * | - | - | - | - | - |
| | 1 | **0.440** | **−0.821** | **0.368** | **−2.234** | **0.026** |
| NewSocial | 0 | - | - | - | - | - |
| | 1 | **1.788** | **0.581** | **0.239** | **2.433** | **0.015** |
| IntrosPeriod*NewSocial | | **3.273** | **1.186** | **0.488** | **2.428** | **0.015** |

**Table A7.** Mature GLMER model parameter estimates for affiliative and submissive response variables. Generalized linear mixed-effect model results of associations between predictor variables (fixed effects) and behavioral response variables for mature focal animals. ß = Beta coefficients from model outputs: positive values indicate an increase in odds compared to the reference level while negative values indicate a decrease in odds compared to reference level; SE = standard error of beta coefficients; Wald $X^2$ = Chi-squared statistic from Wald test with df = 1; $p$ = $p$-value from Wald test. * indicates reference level. Bold signifies statistical significance ($p < 0.05$).

| Predictor | Level | Affiliative | | | | | Submissive | | | | |
|---|---|---|---|---|---|---|---|---|---|---|---|
| | | Odds Ratio | ß | SE | $z$-Statistic | $p$ | Odds Ratio | ß | SE | $z$-Statistic | $p$ |
| SocialAge | Mature * | - | - | - | - | - | - | - | - | - | - |
| | Adolescent | **2.173** | **0.776** | **0.372** | **2.084** | **0.037** | 1.352 | 0.301 | 0.593 | 0.508 | 0.611 |
| | Mixed | 1.185 | 0.170 | 0.471 | 0.360 | 0.719 | 3.029 | 1.108 | 0.853 | 1.300 | 0.194 |
| TimeOfDay | Morning * | - | - | - | - | - | - | - | - | - | - |
| | Afternoon | 0.976 | −0.024 | 0.287 | −0.085 | 0.933 | 0.375 | −0.981 | 0.521 | −1.881 | 0.060 |
| | Night | **0.185** | **−1.688** | **0.229** | **−7.360** | **<0.001** | **0.094** | **−2.365** | **0.463** | **−5.108** | **<0.001** |
| AccessArea | | - | - | - | - | - | **0.954** | **−0.** | **0.022** | **−2.317** | **0.020** |
| Musth | None * | - | - | - | - | - | - | - | - | - | - |
| | Indiv. 2 | **0.431** | **−0.842** | **0.276** | **−3.050** | **0.002** | 0.421 | −0.866 | 0.503 | −1.721 | 0.085 |
| | Indiv. 3 | **0.570** | **−0.562** | **0.228** | **−2.468** | **0.014** | **0.182** | **−1.704** | **0.477** | **−3.575** | **<0.001** |
| | Indiv. 2 and 3 | 0.503 | −0.686 | 0.491 | −1.399 | 0.162 | 1.399 | 0.336 | 0.746 | 0.451 | 0.652 |

**Table A8.** Mature GLMER model parameter estimates for agonistic response variables. Generalized linear mixed-effect model results of associations between predictor variables (fixed effects) and behavioral response variables for mature focal animals. ß = Beta coefficients from model outputs: positive values indicate an increase in odds compared to the reference level while negative values indicate a decrease in odds compared to reference level; SE = standard error of beta coefficients; Wald $X^2$ = Chi-squared statistic from Wald test with df = 1; *p* = *p*-value from Wald test. * indicates reference level. Bold signifies statistical significance (*p* < 0.05).

| Predictor | Level | Non-Contact Agonistic | | | | | Contact Agonistic | | | | |
|---|---|---|---|---|---|---|---|---|---|---|---|
| | | Odds Ratio | ß | SE | z-Statistic | *p* | Odds Ratio | ß | SE | z-Statistic | *p* |
| SocialAge | Mature * | - | - | - | - | - | - | - | - | - | - |
| | Adolescent | 10,449,869 | 16.162 | 9.206 | 1.756 | 0.079 | **5.140** | **1.637** | **0.696** | **2.352** | **0.019** |
| | Mixed | 12,362,769 | 16.330 | 9.214 | 1.772 | 0.076 | **9.854** | **2.288** | **0.911** | **2.510** | **0.012** |
| TimeOfDay | Morning * | - | - | - | - | - | - | - | - | - | - |
| | Afternoon | 1.307 | 0.267 | 0.451 | 0.593 | 0.553 | 0.457 | −0.784 | 0.422 | −1.856 | 0.063 |
| | Night | **0.230** | **−1.481** | **0.450** | **−3.289** | **0.001** | **0.222** | **−1.506** | **0.458** | **−3.290** | **0.001** |
| AccessArea | | - | - | - | - | - | **0.927** | **−0.076** | **0.023** | **−3.292** | **<0.001** |
| InOutAccess | Both * | - | - | - | - | - | - | - | - | - | - |
| | Inside | - | - | - | - | - | **0.157** | **−1.852** | **0.477** | **−3.880** | **<0.001** |
| | Outside | - | - | - | - | - | 0.709 | −0.345 | 0.417 | −0.826 | 0.409 |
| IntrosPeriod | 0 * | - | - | - | - | - | - | - | - | - | - |
| | 1 | **3.002** | **1.099** | **0.375** | **2.932** | **0.003** | **3.639** | **1.292** | **0.327** | **3.950** | **<0.001** |
| NewSocial | 0 * | - | - | - | - | - | - | - | - | - | - |
| | 1 | - | - | - | - | - | **4.546** | **1.514** | **0.311** | **4.868** | **<0.001** |

**Table A9.** Mature GLMER model parameter estimates for proximity response variable. Generalized linear mixed-effect model results of associations between predictor variables (fixed effects) and proximity response variable for mature focal animals. ß = Beta coefficients from model outputs: positive values indicate an increase in odds compared to the reference level while negative values indicate a decrease in odds compared to reference level; SE = standard error of beta coefficients; Wald $X^2$ = Chi-squared statistic from Wald test with df = 1; *p* = *p*-value from Wald test. * indicates reference level. Bold signifies statistical significance (*p* < 0.05).

| Predictor | Level | Proximity | | | | |
|---|---|---|---|---|---|---|
| | | Odds Ratio | ß | SE | z-Statistic | *p* |
| SocialAge | Mature * | - | - | - | - | - |
| | Adolescent | **6.096** | **1.808** | **0.535** | **3.381** | **<0.001** |
| | Mixed | **7.012** | **1.948** | **0.815** | **2.390** | **0.017** |
| TimeOfDay | Morning * | - | - | - | - | - |
| | Afternoon | 0.705 | −0.349 | 0.432 | −0.808 | 0.419 |
| | Night | **0.203** | **−1.595** | **0.434** | **−3.676** | **0.001** |
| InOutAccess | Both * | - | - | - | - | - |
| | Inside | **74.020** | **4.304** | **1.419** | **3.034** | **0.002** |
| | Outside | **283.758** | **5.648** | **1.457** | **3.876** | **<0.001** |
| AccessArea | | **1.301** | **0.263** | **0.078** | **3.388** | **<0.001** |
| Musth | None * | - | - | - | - | - |
| | Indiv. 2 | **0.289** | **−1.243** | **0.372** | **−3.344** | **<0.001** |
| | Indiv. 3 | **0.403** | **−0.908** | **0.330** | **−2.751** | **0.006** |
| | Indiv. 2 and 3 | 0.831 | −0.185 | 0.727 | −0.255 | 0.799 |
| NewSocial | 0 | - | - | - | - | - |
| | 1 | **3.862** | **1.351** | **0.271** | **4.983** | **<0.001** |
| InOutAccess*AccessArea | In | 0.864 | −0.146 | 0.221 | −0.661 | 0.509 |
| InOutAccess*AccessArea | Out | **0.720** | **−0.328** | **0.080** | **−4.114** | **<0.001** |

**Table A10.** New social GLMER model parameter estimates for affiliative and submissive response variables. Generalized linear mixed-effect model results of associations between predictor variables (fixed effects) and behavioral response variables for all focal animals. ß = Beta coefficients from model outputs: positive values indicate an increase in odds compared to the reference level while negative values indicate a decrease in odds compared to reference level; SE = standard error of beta coefficients; Wald $X^2$ = Chi-squared statistic from Wald test with df = 1; $p$ = $p$-value from Wald test. * indicates reference level. Bold signifies statistical significance ($p < 0.05$).

| Predictor | Level | Affiliative | | | | | Submissive | | | | |
|---|---|---|---|---|---|---|---|---|---|---|---|
| | | Odds Ratio | ß | SE | z-Statistic | $p$ | Odds Ratio | ß | SE | z-Statistic | $p$ |
| NewSocial | 0 * | - | - | - | - | - | - | - | - | - | - |
| | 1 | **1.343** | **0.295** | **0.135** | **2.186** | **0.029** | **3.671** | **1.300** | **0.203** | **6.421** | **<0.001** |
| TimeOfDay | Morning * | - | - | - | - | - | - | - | - | - | - |
| | Afternoon | 0.701 | −0.355 | 0.191 | −1.858 | 0.063 | **0.399** | **−0.918** | **0.225** | **−4.082** | **<0.001** |
| | Night | **0.079** | **−2.545** | **0.203** | **−12.555** | **<0.001** | **0.207** | **−2.236** | **0.246** | **−9.071** | **<0.001** |
| InOutAccess | Both * | - | - | - | - | - | - | - | - | - | - |
| | Inside | **0.691** | **−0.370** | **0.154** | **−2.398** | **0.017** | **0.513** | **−0.668** | **0.243** | **−2.747** | **0.006** |
| | Outside | **0.588** | **−0.531** | **0.196** | **−2.711** | **0.007** | 0.749 | −0.289 | 0.229 | −1.260 | 0.208 |
| Musth | None * | - | - | - | - | - | - | - | - | - | - |
| | Indiv. 2 | - | - | - | - | - | **0.621** | **−0.476** | **0.195** | **−2.449** | **0.014** |
| | Indiv. 3 | - | - | - | - | - | **0.626** | **−0.468** | **0.221** | **−2.115** | **0.034** |
| | Indiv. 2 and 3 | - | - | - | - | - | **0.475** | **−0.745** | **0.325** | **−2.294** | **0.022** |
| AccessArea | | - | - | - | - | - | **0.974** | **−0.027** | **0.010** | **−2.697** | **0.007** |
| IntrosPeriod | 0 * | - | - | - | - | - | - | - | - | - | - |
| | 1 | - | - | - | - | - | 1.108 | 0.018 | 0.292 | 0.060 | 0.952 |
| IntrosPeriod*NewSocial | | - | - | - | - | - | 1.931 | 0.658 | 0.349 | 1.885 | 0.059 |

**Table A11.** New social GLMER model parameter estimates for agonistic response variables. Generalized linear mixed-effect model results of associations between predictor variables (fixed effects) and behavioral response variables for all focal animals. ß = Beta coefficients from model outputs: positive values indicate an increase in odds compared to the reference level while negative values indicate a decrease in odds compared to reference level; SE = standard error of beta coefficients; Wald $X^2$ = Chi-squared statistic from Wald test with df = 1; $p$ = $p$-value from Wald test. * indicates reference level. Bold signifies statistical significance ($p < 0.05$).

| Predictor | Level | Non-Contact Agonistic | | | | | Contact Agonistic | | | | |
|---|---|---|---|---|---|---|---|---|---|---|---|
| | | Odds Ratio | ß | SE | z-Statistic | $p$ | Odds Ratio | ß | SE | z-Statistic | $p$ |
| NewSocial | 0 * | - | - | - | - | - | - | - | - | - | - |
| | 1 | **3.516** | **1.257** | **0.370** | **3.397** | **<0.001** | **6.647** | **1.894** | **0.285** | **6.647** | **<0.001** |
| TimeOfDay | Morning * | - | - | - | - | - | - | - | - | - | - |
| | Afternoon | **0.552** | **−0.594** | **0.297** | **−2.002** | **0.045** | **0.443** | **−0.814** | **0.295** | **−2.762** | **0.006** |
| | Night | **0.053** | **−2.931** | **0.318** | **−9.212** | **<0.001** | **0.086** | **−2.455** | **0.327** | **−7.505** | **<0.001** |
| InOutAccess | Both* | - | - | - | - | - | - | - | - | - | - |
| | Inside | - | - | - | - | - | 1.679 | 0.518 | 1.154 | 0.449 | 0.653 |
| | Outside | - | - | - | - | - | 0.467 | −0.762 | 1.096 | −0.695 | 0.487 |
| AccessArea | | - | - | - | - | - | 0.957 | −0.044 | 0.060 | −0.737 | 0.461 |
| Musth | None * | - | - | - | - | - | - | - | - | - | - |
| | Indiv. 2 | 0.742 | −0.298 | 0.282 | −1.058 | 0.290 | 0.639 | −0.447 | 0.260 | −1.717 | 0.086 |
| | Indiv. 3 | 0.573 | −0.556 | 0.352 | −1.580 | 0.114 | **0.410** | **−0.891** | **0.303** | **−2.943** | **0.003** |
| | Indiv. 2 and 3 | 0.446 | −0.809 | 0.451 | −1.792 | 0.073 | 0.922 | −0.081 | 0.417 | −0.195 | 0.846 |
| IntrosPeriod | 0 * | - | - | - | - | - | - | - | - | - | - |
| | 1 | 0.906 | −0.098 | 0.513 | −0.192 | 0.848 | 0.961 | −0.039 | 0.403 | −0.098 | 0.922 |
| InOutAccess*AccessArea | In | - | - | - | - | - | 0.657 | −0.419 | 0.215 | −1.949 | 0.051 |
| InOutAccess*AccessArea | Out | - | - | - | - | - | 1.026 | 0.025 | 0.061 | 0.414 | 0.679 |
| IntrosPeriod*NewSocial | | **3.967** | **1.378** | **0.566** | **2.436** | **0.015** | **3.076** | **1.124** | **0.470** | **2.389** | **0.017** |

**Table A12.** New social GLMER model parameter estimates for proximity response variable. Generalized linear mixed-effect model results of associations between predictor variables (fixed effects) and proximity response variable for all focal animals. ß = Beta coefficients from model outputs: positive values indicate an increase in odds compared to the reference level while negative values indicate a decrease in odds compared to reference level; SE = standard error of beta coefficients; Wald $X^2$ = Chi-squared statistic from Wald test with df = 1; $p$ = $p$-value from Wald test. * indicates reference level. Bold signifies statistical significance ($p < 0.05$).

| Predictor | Level | Proximity | | | | |
|---|---|---|---|---|---|---|
| | | Odds Ratio | ß | SE | z-Statistic | $p$ |
| NewSocial | 0 * | - | - | - | - | - |
| | 1 | **2.097** | **0.741** | **0.239** | **3.094** | **0.002** |
| TimeOfDay | Morning * | - | - | - | - | - |
| | Afternoon | 0.631 | −0.461 | 0.310 | −1.486 | 0.137 |
| | Night | **0.122** | **−2.106** | **0.307** | **−6.869** | **<0.001** |
| InOutAccess | Both * | - | - | - | - | - |
| | Inside | 2.252 | 0.812 | 0.915 | 0.888 | 0.375 |
| | Outside | 3.291 | 1.191 | 0.899 | 1.325 | 0.185 |
| AccessArea | | | | | | |
| Musth | None * | - | - | - | - | - |
| | Indiv. 2 | 0.854 | −0.158 | 0.262 | −0.601 | 0.548 |
| | Indiv. 3 | 0.592 | −0.525 | 0.286 | −1.836 | 0.114 |
| | Indiv. 2 and 3 | 1.024 | −0.024 | 0.413 | 0.057 | 0.955 |
| IntrosPeriod | 0* | - | - | - | - | - |
| | 1 | 0.599 | −0.512 | 0.355 | −1.443 | 0.149 |
| InOutAccess*AccessArea | In | 1.056 | 0.055 | 0.147 | 0.372 | 0.710 |
| InOutAccess*AccessArea | Out | **0.873** | **−0.136** | **0.051** | **−2.665** | **0.008** |
| IntrosPeriod*NewSocial | | 2.265 | 0.818 | 0.476 | 1.717 | 0.086 |

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
