# Peer review of "Age and Social History Impact Social Interactions between Bull Asian Elephants (Elephas maximus) at Denver Zoo"

_2673-5636, doi:10.3390/jzbg4010018_

Round 1

Reviewer 1 Report

The introduction should be shortened. It's too long. Consider one page and a half at most. Remove the paragraph in line 111 to 122. 

Line 179 - How many people rated scan simple? Were they trained for such an assessment? Describe in the methodology.

Line 196 - Were they trained for such an assessment? Which method was used to achieve 95% reliability?

Line 272 - Figure 1, 2 and 3: Figure text needs to be sharper. This color makes it difficult to read.

Line 245-271 -Paragraph too long. Review, you can split in two.

Line 349-351 - How can managers do this?

Line 371-373 - What is your suggestion to avoid this happening again in future experiments? It is interesting to present this in the text.

Line 422-426: I suggest looking for another reference for this behavior, since they are different species with different behaviors.

Reviewer 2 Report

I was very interested in this study and read through it quickly the first time to "see how it ended", truth be told (yes, I know the abstract is like a "spoiler alert" but nonetheless)!  As an "Animal Sociologist" the concept of social history is fascinating to me as are the "sociodemographics" such as age.  I look forward to seeing this work expand.  

Reviewer 3 Report

Very interesting article, rich introduction supported by literature. Professional research workshop. Statistical methods correct. The methodology is described in detail and reliably, however, due to the reviewer's obligation, I point to the following comments:

It is worth presenting tables 1-9 in a more interesting graphical way. Visualizing the results will make them easier to read and make the manuscript more attractive to readers. I leave it to the authors to make the final decision, but I encourage them to consider suggestions for change.

The ratio of the discussion chapter to the introduction seems to be distorted. With such a wealth of literature and analyzed data, one can expect a more detailed discussion of the results. Referring to the hypotheses (123 – 135), there were no isolated conclusions that were a direct response to the assumptions made in the introduction. In my opinion, the statements referring to hypotheses appearing in the discussion chapter are insufficient, and the separation of separate conclusions will be a clear response of the authors to the hypotheses.
